# Efficacy of Whole-Cell-Based Mono- and Bi-Valent Vaccines Against *Nocardia seriolae* and *Aeromonas veronii* in Largemouth Bass, *Micropterus salmoides*

**DOI:** 10.3390/vaccines13090942

**Published:** 2025-09-03

**Authors:** Qiushi Zhang, Nengbin Zhu, Ruiping Xu, Eakapol Wangkahart, Lin Zhang, Lihe Liu, Rui Wang, Zhen Xu, Weiguang Kong, Hongsen Xu

**Affiliations:** 1Hubei Key Laboratory of Animal Nutrition and Feed Science, School of Animal Science and Nutritional Engineering, Wuhan Polytechnic University, Wuhan 430023, China; 2Key Laboratory of Breeding Biotechnology and Sustainable Aquaculture, Institute of Hydrobiology, Chinese Academy of Sciences, Wuhan 430072, China; 3Laboratory of Fish Immunology and Nutrigenomics, Applied Animal and Aquatic Sciences Research Unit, Division of Fisheries, Faculty of Technology, Mahasarakham University, Khamriang Sub-District, Kantarawichai, Mahasarakham 44150, Thailand; 4Yangtze River Fisheries Research Institute, Chinese Academy of Fishery Sciences, Wuhan 430223, China

**Keywords:** *Micropterus salmoides*, *Nocardia seriolae*, *Aeromonas veronii*, Bivalent inactivated vaccine, Immune responses

## Abstract

**Highlights:**

**Abstract:**

Background/Objectives: *Nocardia seriolae* and *Aeromonas veronii* are two important pathogens that can affect a wide range of fish species and cause substantial economic losses. However, a vaccine that simultaneously protects fish from these two bacterial infections is not yet available. Methods: Three formalin-inactivated whole-cell vaccines prepared from *N. seriolae* and *A. veronii* (Monovalent Av, Monovalent Ns and Bivalent Av-Ns) were generated, and their efficacy was evaluated through a range of tests. The immune-related gene expression in the spleen and head kidney, enzyme activity, and specific antibody levels in serum were also detected. Results: All groups of vaccinated fish exhibited increased serum enzymatic activity compared with control fish, which peaked at week 3 after vaccination; in particular, that of the Bivalent Av-Ns group increased remarkably. The expression of immune-related genes in the spleen, head, and kidneys increased after immunization and were significantly enhanced (*p* < 0.05) in the bivalent vaccine group. Specific antibodies were produced at the 1st wpv, peaked at the 4th to the 5th wpv, and then decreased at the 6th wpv in all vaccinated groups. The Monovalent Av and Monovalent Ns against *A. veronii* and *N. seriolae* showed 56.67% and 22.22% RPS, respectively. Moreover, Bivalent Av-Ns offered 33.33% and 76.67% RPS for single infection with *N. seriolae* or *A. veronii*, as well as providing 44.44% RPS for dual infection with combined *N. seriolae* and *A. veronii*. Conclusions: Our findings indicate that the administration of the *A. veronii* and *N. seriolae* bivalent vaccine can protect largemouth bass from both bacterial infections.

## 1. Introduction

Largemouth bass (*Micropterus salmoides*) is one of the most important freshwater fish widely cultured in China [1]. It is favored by consumers and farmers because of its fast growth rate, delicious taste, and high economic potential, as well as its high proportion of protein and omega-3 fatty acids [2]. Native to southeastern America, the fish was first introduced into China in the 1980s and became a prominent freshwater fish species [3]. China has established itself as the global leader in largemouth bass production, with an annual output exceeding 938 thousand tons as of 2024 [4]. However, due to being reared in crowded and stressful conditions, largemouth bass have become more susceptible to infectious diseases, which have plagued their production [5]. One of the main factors limiting the sustainable development of the largemouth bass farming industry is the appearance of pathogenic bacteria, such as *Aeromonas veronii* [6], *Flavobacterium columnare* [7], *Edwardsiella piscicida* [8], *Streptococcus iniae* [9], *Plesiomonas shigelloides* [10], and *Nocardia seriolae* [11].

*N. seriolae* and virulent *A. veronii*, two common bacteria that are found widely in the aquaculture industry, have been reported to induce nocardiosis and enteric septicemia disease, respectively, in largemouth bass [12]. *A. veronii* is a Gram-negative bacterium that is generally widespread in environments such as soil, water, and plant surfaces [13]. Typical clinical signs of *A. veronii*-infected fish include dermal ulceration, furunculosis, enteritis, and hemorrhagic septicemia [14]. Over the last decade, an increasing number of disease outbreaks caused by *A. veronii* infection have been documented in various aquatic animals, such as the Nile tilapia (*Oreochromis niloticus* L.) [15], European seabass (*Dicentrachus labrax*) [16], loach (*Misgurnus anguillicaudatus*) [17], and crucian carp (*Carassius auratus gibelio*) [18]. *N. seriolae*, a pathogen belonging to the Nocardiaceae family, is a Gram-positive, pleomorphic acid-fast, branching, filamentous intracellular bacterium that can cause fish nocardiosis [19]. In recent years, the incidence of fish nocardiosis has been increasing yearly, causing huge commercial losses in the global aquaculture industry, especially in Asia [20]. Aquatic animals infected by this bacterium are characterized by the chronic progression of symptoms such as skin ulcers and granulomas in the kidney, liver, muscle, and spleen tissues [21]. To date, fish nocardiosis has been reported in many fish species, including yellowtail (*Seriola quinqueradiata*) [22], Japanese sea perch (*Lateolabrax japonicus*) [23], jade perch (*Scortum barcoo*) [24], and snakehead (*Channa argus*) [25]. In our previous research, largemouth bass were found to be simultaneously infected with these bacteria [26], leading to greater concern regarding co-infection with those two bacteria in fish farming.

Traditionally, the application of antibiotics and chemicals for bacterial disease prevention and treatment is effective, but the overuse of these agents can result in adverse side effects, including the appearance of drug-resistant bacteria, antibiotic accumulation in water and aquatic organisms, and growing concern for human health and food safety [27]. Vaccination is considered as one of the most effective and acceptable methods for preventing disease outbreaks in the aquaculture industry [28]. Furthermore, vaccine administration assists in initiating a long-lasting adaptive immune response against microorganisms. Therefore, some vaccines have been experimentally investigated and even commercially implemented to protect fish of economic importance [29,30]. However, most of the vaccines that have been reported are monovalent formats and protect against a single pathogen disease. Compared with one single inactivated vaccine, the use of bivalent inactivated vaccines can not only protect the host against two pathogens at the same time, reducing the number of doses required, but also improve the efficiency and convenience of the vaccination process [31]. However, research on the development of a bivalent vaccine for controlling *A. veronii* and *N. seriolae* infection in largemouth bass is not yet available, which poses a significant challenge in the fish industry [32].

In this study, whole-cell-based monovalent and bivalent vaccines containing *A. veronii* and *N. seriolae* were prepared, and their effectiveness in terms of immune response and disease prevention in largemouth bass was investigated. This represents a practical method of protecting largemouth bass from these two bacterial infections and shows potential for effective application in aquaculture farming in the future.

## 2. Materials and Methods

### 2.1. Ethics Statement

This research was conducted strictly within the ethical standards and the guidelines of “Regulations for the Administration of Affairs Concerning Experimental Animals” documented by the State Science and Technology Commission of Hubei Province. Additionally, all the experimental methods used in this study were also approved by Wuhan Polytechnic University’s Department of Risk Management and Safety (Reference number: WPU202212002; Approval date: 21 March 2022).

### 2.2. Fish Husbandry

Healthy largemouth bass (35 ± 5 g) were obtained from a local fish hatchery in Wuhan, Hubei province, China. Prior to vaccination, the fish underwent a two-week acclimation period in 5000 L tanks with continuous aeration. They were fed a commercial diet (Yuehai Feed Company Co., Ltd., Zhanjiang, China) at 2% of their body weight twice daily and were confirmed to be pathogen-free, as described previously [33]. Water quality parameters (temperature, 30.0 ± 1.0 °C; pH, 7.00 ± 0.20; ammonia-N, 0.03 ± 0.01 mg/L; dissolved oxygen, 7.75 ± 0.25 mg/L) were checked and maintained during the experimental period.

### 2.3. Preparation of Formalin-Inactivated Mono- and Bi-Valent Vaccines

The pathogenic strains of *A. veronii* 21AV928 and *N. seriolae* 21NS928 were previously isolated from naturally diseased largemouth bass and exhibited strong pathogenicity against fish [26]. The pathogenic *A. veronii* and *N. seriolae* were inoculated in Luria–Bertani (LB) and Brain Heart Infusion (BHI) broth with constant shaking at 28 °C for 24 h and 5 days, respectively. Then, each bacterium was collected via centrifugation at 8000× *g* for 10 min. After being washed three times with sterile PBS, the bacterial solutions were diluted to 2 × 10^9^ CFU/mL. The inactivation of *A. veronii* and *N. seriolae* was achieved via incubation with 0.3% (*v*/*v*) formaldehyde at 28 °C for 1 day. After inactivation, the bacterial cells were collected via centrifugation at 8000× *g* for 10 min. Subsequently, the bacterial precipitation was washed with sterile PBS three times. For the preparation of the bivalent vaccine, suspensions of *A. veronii* and *N. seriolae* were mixed in equal volumes to a final concentration of 2 × 10^9^ CFU/mL and then inactivated as previously mentioned. Sterility was confirmed as a lack of bacterial growth after the inoculation of 0.1 mL of inactivated bacterial cells on LB and BHI agar plates and incubation at 28 °C for 96 h and 10 days, respectively. The safety of inactivated vaccines was confirmed via intraperitoneal injection into healthy largemouth bass at 100 μL/fish (1.0 × 10^8^ CFU/mL) and the observation of no abnormal signs or mortality over the two weeks post injection. The inactivated bacterins were stored at 4 °C before usage.

### 2.4. Vaccination and Sample Collection

Healthy largemouth bass were randomly distributed into four treatment groups (*n* = 180 per group). Each group was cultured in freshwater in three circular tanks (200 L capacity) with 60 fish per tank and intraperitoneally administrated with 200 μL of PBS (control), monovalent inactivated *A. veronii* (Monovalent Av), monovalent inactivated *N. seriolae* (Monovalent Ns), or bivalent inactivated *A. veronii* and *N. seriolae* (Bivalent Av-Ns). To investigate the level of immune-related gene transcription, three fish from each group were dissected at 12 h, 24 h, 48 h, 72 h, and 96 h post vaccination (hpv), and tissues from the spleen and head kidney were collected and immediately preserved in RNA (TaKaRa, Dalian, China) for subsequent RNA extraction. For serum isolation, three fish per group were randomly collected at the 1st, 2nd, 3rd, 4th, 5th, and 6th weeks post vaccination (wpv), and blood was sampled from the caudal vein with a 1 mL syringe. After clotting at 28 °C for 1 h, the serum was obtained via centrifugation at 4000× *g* for 5 min and then stored at −80 °C for the future detection of specific antibody titers and non-specific enzyme activities.

### 2.5. Measurement of Serum Immune Parameters

After thawing, serum samples were selected to determine the activities of acid phosphatase (ACP; Cat. No. A060–2), alkaline phosphatase (AKP; Cat. No. A059–2), and lysozyme (LZM; Cat. No. A050) as per the manufacturer’s instructions for the commercial kits produced by Nanjing Jiancheng Bioengineer Institute (Nanjing, China).

### 2.6. Enzyme-Linked Immunosorbent Assay (ELISA)

Serum samples were heat-inactivated at 56 °C for 30 min and used for specific antibody titers against *A. veronii* and *N. seriolae* via ELISA as previously reported [34]. In brief, 96-well ELISA microtiter plates (Costar, Cambridge, MA, USA) were coated with 0.1 mL *A. veronii* and *N. seriolae* bacterial suspensions (1.0 × 10^8^ CFU/mL) at 4 °C overnight. Following a 1 h blocking step with 3% BSA at 37 °C, 0.1 mL of anti-serum (1:200) from vaccinated largemouth bass was added to each well and incubated at 37 °C for 1 h. After that, the wells were incubated with 0.1 mL of rabbit polyclonal antibodies against largemouth bass IgM (1:2000 dilution, kindly provided by Dr. Shun Yang, Laboratory of Life Science and Medicine, Zhejiang Sci-Tech University) [35] at 37 °C for 50 min. Then, 0.1 mL of horseradish peroxidase conjugated goat-anti-rabbit IgG (1:5000) was added into each well and incubated in 37 °C for 45 min. Between each step, the wells of the microplates were washed 3 times with PBST (PBS containing 0.1% Tween-20). The color reaction was developed with the TMB kit (Solarbio, Beijing, China) and stopped by adding 0.1 mL/well of 1 M HCl. After that, the absorbance was measured at OD450 using a Precision Microplate Reader (Thermo Scientific, Waltham, MA, USA). As a negative control, serum from immunized fish was replaced with 0.1 mL of serum from PBS-injected fish.

### 2.7. Quantitative Real-Time PCR (qRT-PCR)

The transcriptional profiles of immune-associated genes in the spleen and head kidney were analyzed via qRT-PCR [36]. Briefly, the total RNA in the tissues was extracted using RNAiso Plus reagent (TaKaRa Bio Inc., China), and its quality was examined via 1.0% gel electrophoresis. Meanwhile, the RNA concentration was measured based on the A260/280 absorbance ratio using a microspectrophotometer (Beijing Kaiao, China) and then diluted to 0.1 μg/μL with ddH_2_O. Afterwards, cDNA was synthesized with the PrimeScript^TM^ RT reagent kit (TaKaRa, Kusatsu, Japan) as per the manufacturer’s protocols. The qRT-PCR for test genes was carried out with an SYBR^®^ Premix Ex Taq™ II Kit (Takara, China) and performed using an ABI 7500 Real-Time PCR system (Applied Biosystems, USA) with a reaction volume of 20 μL, containing 2 × Premix Ex Taq (Takara, China), with 0.4 μL of each primer (10 μM), 6.8 μL of H_2_O, and 1 μL of the DNA template. The thermal cycling protocol was as follows: initial denaturation for 10 min at 95 °C, followed by 40 cycles of 95 °C for 10 s and 60 °C for 30 s. Melting curves were generated by monitoring the fluorescence from 65 °C to 95 °C. The relative expression of target genes was analyzed using the 2^−ΔΔCT^ method [37], with β-actin serving as the housekeeping gene based on previous studies [38,39,40]. All primers were provided by Tsingke (Wuhan, China), and their sequences are listed in Table 1.

### 2.8. Challenge Test

At 6 wpv, out of the remaining fish in the Monovalent Ns and Monovalent Av groups, 30 were randomly selected per group and independently injected with 0.2 mL of *A. veronii* (3.18 × 10^8^ CFU/mL) and *N. seriolae* (1.26 × 10^9^ CFU/mL), respectively. Meanwhile, fish from the control and bivalent inactivated vaccine groups were randomly distributed into three subgroups (30 fish/subgroup) and then challenged separately with 200 μL virulent strains of *A. veronii* (3.18 × 10^8^ CFU/mL) and *N. seriolae* (1.26 × 10^9^ CFU/mL) and combined bacteria (containing 3.18 × 10^8^ CFU/mL of *A. veronii* and 1.26 × 10^9^ CFU/mL of *N. seriolae*). The mortality was monitored over a 14-day period post challenge, and the relative percent survival (RPS) was calculated as follows: RPS = {1 − (%mortality in vaccinated fish/% mortality in control fish)} × 100% [33].

### 2.9. Statistical Analysis

All data are expressed as the mean ± standard deviation (SD) of three experimental replicates. Differences between groups were analyzed for statical significance using one-way analysis of variance (ANOVA) with a post hoc Duncan’s multiple range test (DMRT) in SPSS 26.0 software (SPSS Inc., Chicago IL, USA). Data were defined as statistically significant at *p* < 0.05.

## 3. Results

### 3.1. Analysis of Serum Non-Specific Immune Parameters

As presented in Figure 1, the activities of LZM, ACP, and AKP in all vaccinated groups first increased, reaching their peak value between the first and third wpv, and subsequently decreased at the end of the trial. In comparison, the LZM, ACP, and AKP activities of the control group fluctuated slightly during the experimental period, with changes that were not significant (*p* > 0.05).

The activity of ACP was remarkably improved at the 1st wpv (*p* < 0.05), peaked at the 3rd wpv, and then gradually declined in both the Monovalent Ns and Monovalent Av group. The ACP activity showed remarkable enhancement at the 1st wpv (*p* < 0.05) and reached its maximum value at the 2nd wpv, followed by a gradual decrease in the Bivalent Av-Ns group (Figure 1A).

The AKP activity was significantly (*p* < 0.05) elevated at the 1st wpv and reached its peak value at the 2nd wpv in the Bivalent Av-Ns group. Meanwhile, the AKP activity was obviously increased at the 1st wpv (*p* < 0.05) and reached its peak at the 3rd wpv, followed by a stable descent in both the Monovalent Ns and Monovalent Av group (Figure 1B).

As presented in Figure 1C, the LZM activities in all vaccinated groups were remarkably (*p* < 0.05) higher than those in the control group. At the 3rd wpv, the LZM activities in both the Monovalent Ns and Monovalent Av groups reached their highest level, which significantly exceeded the level of the control group, and the highest LZM activities were 206.7 ± 1.7 µg/mL and 185.2 ± 1.4 µg/mL, respectively. The LZM activity in the serum of the bivalent inactivated vaccine group had increased rapidly (*p* < 0.05) at the first wpv, then reached its peak at the 2nd wpv (172.4 ± 1.2 μg/mL), maintained the higher level at the 3rd wpv (170.4 ± 1.1 μg/mL), and exceeded the control group until the 6th wpv.

### 3.2. Measurement of Serum Specific Antibody Titers

The specific antibody titers in largemouth bass against *N. seriolae* or *A. veronii* were measured using the ELISA method at the 1st, 2nd, 3rd, 4th, 5th, and 6th wpv. As shown in Figure 2, in the sera of the Monovalent Av, Monovalent Ns and Bivalent Av-Ns groups, the specific antibodies against both *A. veronii* and *N. seriolae* showed a similar tendency of first rising, reaching their peak level at the 4th or 5th wpv, and then decreasing at the 6th wpv. Meanwhile, the antibody titers against *N. seriolae* and *A. veronii* in the control group remained relatively unchanged during the experiment.

As shown in Figure 2A, the Monovalent Av and Bivalent Av-Ns groups showed a higher antibody titer against *A. veronii* than the control group from the 1st to the 6th wpv. Antibodies against *A. veronii* showed a prominent increase in the Bivalent Av-Ns group over the entire experimental period, reaching their peak at the 4th wpv (*p* < 0.05) and maintaining high levels at the 5th and 6th wpv. In the Monovalent Av group, the antibody titer against *A. veronii* improved obviously (*p* < 0.05) and reached its peak level at the 6th wpv. As shown in Figure 2B, the values of specific antibodies against *N. seriolae* exhibited an obvious increasing trend at the 1st wpv in both the Monovalent Ns and Bivalent Av-Ns groups, peaked at the 5th wpv, and then declined stably.

### 3.3. The Expression of Immune-Related Genes in the Spleen Following Vaccination

The expression of genes related to immune response in the spleen tissue was determined at 12, 24, 48, 72, and 96 hpv via qRT-PCR. As shown in Figure 3, the transcription profiles of tested genes in all vaccine groups showed a similar tendency to first increase, reach their peak values from 12 to 72 hpv, and then gradually decrease at the end of the experiment. The *IL-8* and *IL-10* gene transcription profiles in the Monovalent Av, Monovalent Ns, and Bivalent Av-Ns groups were significantly upregulated and reached their peak values 24 h after immunization, whereas the mRNA levels of the *CD8α*, *IL1β*, and *IgT* genes increased significantly and reached their maxima at 48 hpv in all the vaccinated groups. As for *CD4-1* and *TNF-α*, the highest gene expression levels were obtained at 24 hpv in the Bivalent Av-Ns group, while the maximum levels were detected at 48 hpv in the Monovalent Av and Monovalent Ns groups. The highest fold changes in the *MHCI-α* gene were observed at 12 hpv, 48 hpv, and 72 hpv in the Bivalent Av-Ns, Monovalent Ns, and Monovalent Av group, respectively. The expression levels of *IgM* and *MHCII-α* both reached peak levels at 12 hpv, prior to those in the monovalent groups. In the control group, all tested gene expression profiles remained relatively unchanged throughout the experimental period.

### 3.4. The Expression of Immune-Related Genes in the Head Kidney After Vaccination

qRT-PCR was employed to detect the expression profiles of immune-associated genes in the head kidney of largemouth bass, and the results are displayed in Figure 4. Compared to the control group, the transcription values for all detected genes in the head kidney were notably increased after immunization with three inactivated vaccines. The mRNA expression profiles of the *MHCI-α*, *IgM*, *IL-1β*, and *IL-8* genes in all vaccine groups were higher than those in the control group, reaching the highest level at 24 hpv and then decreasing. As for *CD4-1*, *IL10*, and *CD8α*, the highest gene expression levels were obtained at 24 hpv in the Monovalent Av and Bivalent Av-Ns groups, while the maximum level was detected at 48 hpv in the Monovalent Ns group. The highest fold changes in the *IFN-γ*, *TNF-α*, *MHCII-α*, and *IgT* genes were observed in the Monovalent Av and Bivalent Av-Ns groups at 12 hpv, whereas the transcript of these genes reached its peak level at 24 hpv in the Monovalent Ns group.

### 3.5. Immune Protective Effects

At 6 weeks post vaccination, the immunized fish were intraperitoneally challenged with *N. seriolae* or *A. veronii* and monitored for mortality over 2 weeks. The results demonstrated that the RPS rates in all vaccinated groups remarkably improved after immunization (Figure 5). Specifically, after challenge with the pathogenic *A. veronii*, the RPS of largemouth bass in the Monovalent Av and Bivalent Av-Ns groups was 56.67% and 76.67%, respectively (Figure 5A), while after challenge with the pathogenic *N. seriolae*, the RPS of largemouth bass in the Monovalent Ns and Bivalent Av-Ns groups was 22.22% and 33.33%, respectively, at 14 d post challenge (Figure 5B). Moreover, after co-challenge with the pathogenic *A. veronii* and *N. seriolae,* the RPS of largemouth bass in the Bivalent Av-Ns groups was 44.44% (Figure 5C).

## 4. Discussion

Recently, outbreaks of nocardiosis and enteric septicemia disease have resulted in high mortality in the intensive farming of largemouth bass, seriously impeding the healthy and sustainable development of the aquaculture industry [6,11]. Moreover, largemouth bass either mono-infected or co-infected with *A. veronii* and *N. seriolae* have been reported [6,11,26], highlighting the urgent need for an efficient prophylactic measure to control those two bacterial infections. Vaccinations containing antigens can stimulate immune responses and have emerged as a predominant strategy for preventing diseases in aquaculture [41]. Monovalent vaccines containing a single bacterium have often been used to prevent diseases in the aquaculture industry [42]; however, vaccination via certain routes, such as intraperitoneal injection, requires the fish to be inoculated more than once to achieve effective immunization, particularly against concurrent infections. This will lead to a stress response in the animal and increase vaccination costs. Therefore, polyvalent vaccines have emerged as the optimal strategy to combat disease in fish farming because they offer advantages including high safety, time savings, and cost-effectiveness [27]. Here, we successfully developed whole-cell-based mono- and bivalent vaccines against *A. veronii* and *N. seriolae* infection and then tested their immunological and protective efficacy in largemouth bass when administrated via intraperitoneal injection.

It is well known that innate immunity is immediately triggered when antigens enter the body. After vaccination, the activities of several immune-associated enzymes, including LZM, AKP, and ACP, were measured in serum. LZM is an important constituent of the innate immune response and plays a vital role in controlling bacterial invasion by activating the complement system and lysing pathogens [43]. In this study, the LZM activity in the vaccinated group was remarkably increased from the 1st wpv, reached a peak level at the 2nd–3rd wpv, and maintained a high value until the 6th wpv. Similar results have also been found in turbot following vaccination with a bivalent vaccine consisting of *A. salmonicida* and *E. tarda* [27], as well as hybrid red tilapia after immunization with a bivalent vaccine targeting *S. iniae* and *A. hydrophila* [44]. ACP and AKP are lysosomal enzymes that play a critical role in innate immunity in fish by breaking down pathogens, promoting the immune response, and resisting invasive organisms [45]. Moreover, their activities have been regarded as indicators of macrophage activation, reflecting the possibility of the intracellular digestion of engulfed antigens [46]. In this research, we demonstrated a significant increase in AKP and ACP activities in vaccinated largemouth bass compared to control fish. Meanwhile, compared to that fish that received single *A. veronii* or *N. seriolae* immunization, these activities were increased significantly and maintained for longer in fish immunized with the *A. veronii* and *N. seriolae* bivalent vaccine. This result is in line with previous research demonstrating a similar trend in ACP and AKP activities after immunization with a bivalent inactivated vaccine containing *A. veronii* and *E. ictaluri* [47]. Generally, the upregulation of LZM, ACP, and AKP activities in the serum implies a robust activation of the innate immune system following vaccination in largemouth bass.

Upon antigenic stimulation, the host elicits a series of complex innate immune responses, including phagocytosis, the secretion of inflammatory mediators, and cell recruitment, all of which are fundamental in triggering a robust antigen-specific adaptive immune response [48]. *TNF-α*, *IL-1β*, and *IL-8*, three key pro-inflammatory cytokines, play a vital role in suppressing the inflammatory response and maintaining immune balance [49]. As expected, the transcript values of *IL-8*, *IL-1β*, and *TNF-α* were remarkably improved post vaccination, implying their crucial role in regulating the immune response post vaccination. Endogenous antigenic peptides are exposed to CD8^+^ T lymphocytes by MHC I molecules, whereas MHC II molecules expose exogenous proteins to helper CD4^+^ T cells [50]. In this research, an obvious increase was detected in the transcriptional levels of genes associated with antigen presentation (*MHC I* and *MHC II*) and T cell markers (*CD4-1* and *CD8α*) in spleen and head kidney tissues, implying that both monovalent and bivalent vaccines contribute to stimulating MHC I-CD8 and MHC II-CD4 antigen presentation pathways in largemouth bass. These findings are in line with a previous study that found that MHC I and MHC II pathways were both initiated post vaccination [30]. During a pathogen invasion, *IL 10* and *IFN-γ* immunomodulators play a role in suppressing the immune response, protecting the host from excessive inflammation and immune response damage, and regulating the activity of immune cells as well as maintaining balance in the immune system [51]. Zhang et al. also showed that zebrafish immunized with a *Vibrio anguillarum* vaccine showed a high expression profile for *IL-10* [52]. These results revealed that the vaccine could stimulate both a pro-inflammatory and anti-inflammatory reaction at the same time, thereby maintaining immune homeostasis and protecting tissue from damage caused by inflammatory responses. IgM and IgT immunoglobulins produced by B lymphocytes are a key component of the immune system. They help the host fight infection by recognizing, binding, and neutralizing pathogens [53]. In our study, in comparison with the control group, the transcript levels of immune-related genes were evaluated after immunization with monovalent and bivalent inactivated vaccines, showing that the immune response was activated after immunization. Similar results were also observed in a prior investigation, which reported an obvious increase in immune-associated gene expression in the spleen and kidney of red hybrid tilapia following the oral administration of bivalent vaccines containing *Streptococcus* and *Aeromonas* [54]. Furthermore, we found that a bivalent inactivated vaccine significantly enhanced the transcript levels of test genes compared to those of monovalent vaccine groups, suggesting that the mixed antigens induce a stronger immune response.

As we know, successful vaccination depends on success in improving immunological memory, in order to respond swiftly and effectively to a subsequent infection by the same specific microbes [55]. This can be accomplished through the previous production of long-lived memory T and B lymphocytes during the primary immune response [56]. In the present research, we immunized largemouth bass and subsequently challenged them with related bacteria at the 6th wpv. The results showed that high protection was achieved in vaccinated fish compared to those from the control group. This is perhaps due to the production of immunological memory that helps fish defend against a subsequent re-encounter with the target pathogen. Previous research has found that the antibody levels of fish that had been immunized with an inactivated vaccine over 8 weeks increased immediately after challenge within 1 week [57]. Many factors, such as the vaccination method, initial antigen dose, and antigen type may affect the formation of immune memory [58]. Tilapia intraperitoneally immunized with an inactivated vaccines showed a significantly higher RPS than those vaccinated via the intramuscular route [59]. This difference was due to the fact that the bioavailability of the antigens via the intraperitoneal route was 4.5 times higher than that via the intramuscular route [60]. The dose dependency of memory formation was studied in detail in carp, and it was found that fish immunized with 10^9^ sheep red blood cells showed high levels of secondary response [61]. Previous research showed that specificity and immunological memory were activated using the *Edwardsiella tarda* vaccine (10^9^ CFU/mL), as demonstrated by the increase in antibody titers and the percentage of CD4^+^ T and CD8^+^ T lymphocytes [62]. In our study, 10^9^ CFU/mL formalin-killed bacteria were selected as the target vaccine for immunization.

The antibody titer is an important indicator for evaluating vaccine efficiency [63]. After vaccination, the host rapidly produce IgM antibodies, which can quickly recognize and neutralize pathogens, providing early protection [64]. Moreover, the IgM produced also functions by activating antigen-specific B cells and enhancing phagocytosis [65]. Specific antibody titers in the serum of the vaccine groups significantly exceeded that in the control group when tested for both bacterial antigens. These results were in line with previous research that showed that the antibody levels against *V. anguillarum* and *E. piscicida* in both monovalent- and bivalent-vaccinated turbot clearly improved compared to unvaccinated fish [66]. Likewise, Md Shirajum et al. observed that tilapia immunized with a bivalent *S. iniae* and *A. hydrophila* vaccine produced significantly higher-value antibody titers against both bacteria [44]. This phenomenon highlights the initiation of B cell-based immunity after vaccination.

The RPS has always been regarded as a direct indicator for evaluating vaccine efficiency. High levels of RPS in vaccinated fish are often associated with the activation of the humoral immune response followed by an improvement in the innate immune system [67]. Our results indicate that the RPS was significantly higher in all vaccine groups compare to the control groups. In addition, the RPS of the Bivalent Av-Ns group was significantly increased compared with the Monovalent Ns and Monovalent Av groups after being challenged with *A. veronii* and *N. seriolae*, respectively. These results demonstrated that the bivalent vaccines conferred a higher RPS than the monovalent vaccines, similarly to Md Shirajum’s results, which showed that a bivalent inactivated vaccine afforded high protection in comparison with a monovalent vaccine [44]. It is also worth noting that Monovalent Ns conferred relatively low protection against *N. seriolae* (RPS = 22.22%) compared to Monovalent Av against *A. veronii* infection (RPS = 56.67%). We speculate that the inactivated *N. seriolae* vaccine mainly increases the humoral immune response, which might contribute to a limited role in defense against *N. seriolae.* In addition, the inactivated bivalent vaccine induced good protection against *A. veronii* and *N. seriolae* co-infection (RPS = 44.44%). Our findings indicate that the bivalent inactivated vaccine represents an effective strategy to protect fish against *A. veronii* and *N. seriolae* mono- or co-infection.

## 5. Conclusions

In summary, whole-cell-based mono- and bivalent vaccines against *N. seriolae* and *A. veronii* were developed to protect largemouth bass against these two bacterial infections. Enhancements in *N. seriolae*- and *A. veronii*-specific antibody titers, enzyme activities, and the expression of immune-related genes further confirmed the stimulation of immunity in largemouth bass. These results indicate that the bivalent vaccine including *N. seriolae* and *A. veronii* is a promising candidate for protecting largemouth bass against nocardiosis and enteric septicemia disease.

## Figures and Tables

**Figure 1 vaccines-13-00942-f001:**
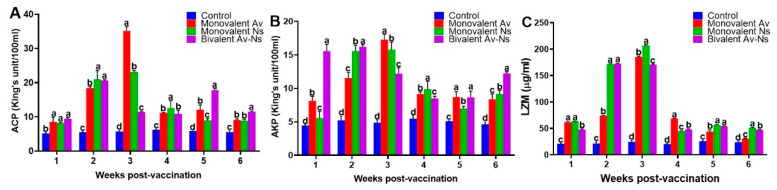
Enzyme activity in sera of vaccinated largemouth bass. (**A**) ACP; (**B**) AKP; (**C**) LZM. Different letters above the bars show significant diversity (*p* < 0.05) among the control and vaccine groups at each time point.

**Figure 2 vaccines-13-00942-f002:**
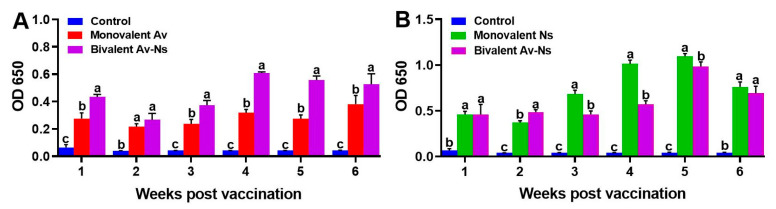
Specific antibody titers in sera of largemouth bass post immunization. (**A**) Specific antibody titers against *Aeromonas veronii*; (**B**) specific antibody titers against *Nocardia seriolae*. Different letters above the bars show significant diversity (*p* < 0.05) among control and vaccine groups at each time point.

**Figure 3 vaccines-13-00942-f003:**
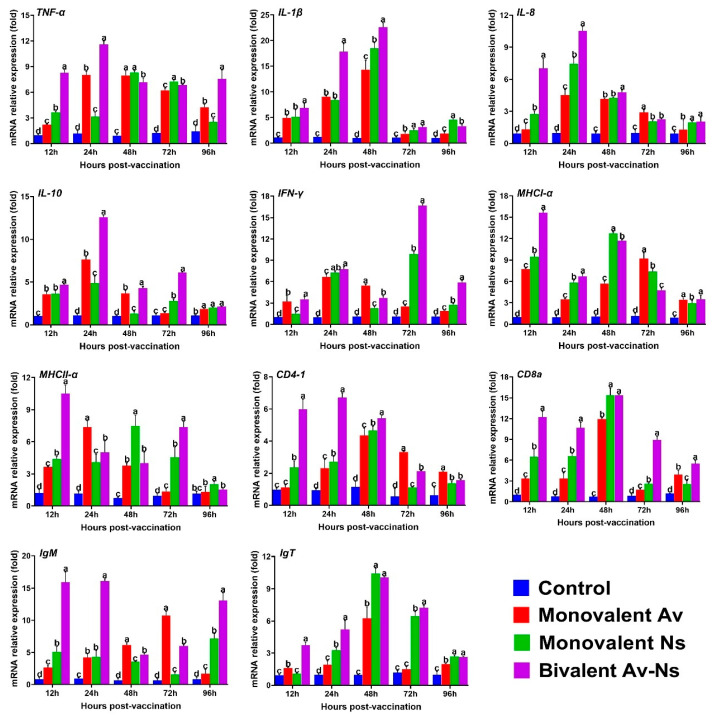
The expression profiles of immune related genes in the spleen post immunization. All values are expressed as the mean ± SD (*n* = 3), and different letters above the bars indicate statistically significant differences among different groups at the same time point (*p* < 0.05).

**Figure 4 vaccines-13-00942-f004:**
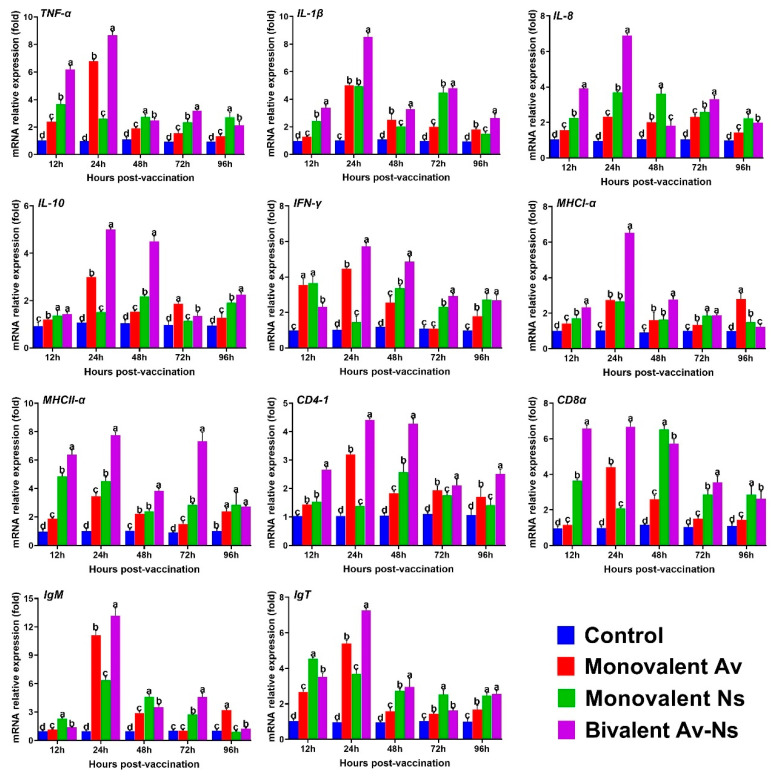
The expression levels of immune-related genes in the head kidney after immunization. Different letters above the bars indicate statistically significant differences among different groups at the same time point (*p* < 0.05).

**Figure 5 vaccines-13-00942-f005:**
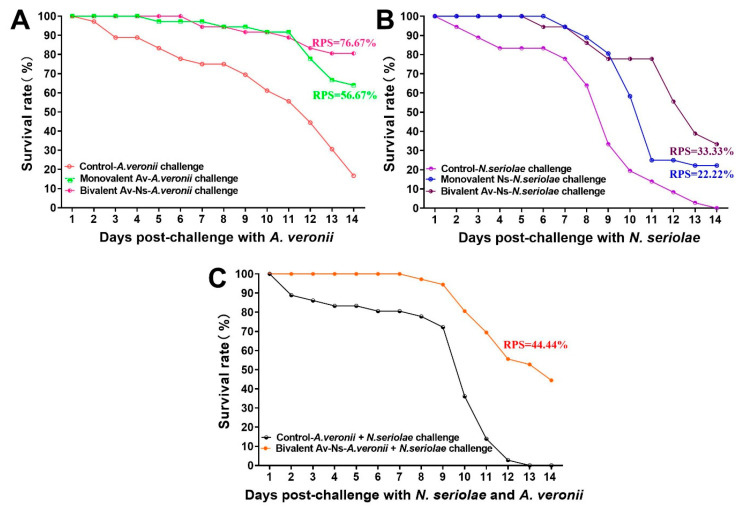
The survival rate and RPS of immunized largemouth bass after mono- or co-infection with *Aeromonas veronii* and *Nocardia seriolae*. (**A**) Mono-infection with *A. veronii*; (**B**) Mono-infection with *N. seriolae*; (**C**) co-infection with *A. veronii* and *N. seriolae*.

**Table 1 vaccines-13-00942-t001:** Primer sequences used in qRT-PCR assay.

Target Gene	F: Primer Sequence (5′-3′)	R: Primer Sequence (5′-3′)	Amplicon Size (bp)	Accession No.	Reference
*TNF-α*	CTTCGTCTACAGCCAGGCATCG	TTTGGCACACCGACCTCACC	161	XM_038710731.1	[38]
*IL-1β*	CGTGACTGACAGCAAAAAGAGG	GATGCCCAGAGCCACAGTTC	166	XM_046034892.1	[38]
*IL-8*	CGTTGAACAGACTGGGAGAGATG	AGTGGGATGGCTTCATTATCTTGT	107	MW751832.1	[38]
*IL-10*	ACAACCAGTGCTGCCGTT	GCAGCGCTGTGTCTAAGTCA	117	XM_038696252.1	[39]
*IFN-γ*	TGCAGGCTCTCAAACACATC	TGTTTTCGGTCAGTGTGCTC	105	XM_038707474.1	[40]
*CD4-1*	GCTCCAGCGGGGAATAATTT	GCCAGGCAAGCTCAAAGTTA	73	XM_038711093.1	[40]
*CD8* *α*	GGAAGGGGATCCTGTTGACA	CCAGCACTCGAAACCAGATG	74	XM_038696403.1	[40]
*IgM*	CTGGACCAGTCTCCCTCTGA	CGAGGTACTGAGTGCTGCTG	235	MZ396108.1	[40]
*IgT*	AAAGGAGATGGGAGTGAGCC	GTTGGGTCTTCTGTGGGGG	199	MZ388129.1	[40]
*MHCI-α*	GTGGTTCAACGTCAACATCG	ACCCAGACTTGTTCGGTGTC	198	XM_046058860.1	[40]
*MHCII-α*	GAGGACCTTGCTGTCATTGG	GCGTACCAAACCTCTTCACC	98	XM038696307	[40]
*β-actin*	CCACCACAGCCGAGAGGGAA	TCATGGTGGATGGGGCCAGG	303	MH018565.1	[40]

Note: *TNF-α*, tumor necrosis factor-α; *IL-1β*, interleukin-1β; *IL-8*, interleukin-8; *IL-10*, interleukin-10; *IFN-γ*, interferon-γ; *CD4-1*, cluster of differentiation 4-1; *CD8α*, cluster of differentiation 8a; *IgM*, immunoglobulin M; *IgT*, immunoglobulin T; *MHC I-α*, major histocompatibility complex I-α; *MHC II-α*, major histocompatibility complex II-α; *β-actin*, beta-actin.

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
