# Peer review of "Efficacy of Whole-Cell-Based Mono- and Bi-Valent Vaccines Against Nocardia seriolae and Aeromonas veronii in Largemouth Bass, Micropterus salmoides"

_vaccines, 2025, doi:10.3390/vaccines13090942_

Round 1

Reviewer 1 Report

Comments and Suggestions for Authors

The manuscript focuses on fully filling the knowledge gap and the need for bivalent vaccines to protect against N. seriolae and A. veronii. Results are conclusive and well presented. The aim of the study is well addressed. The authors did a complete assessment of innate and adaptive immune responses. Even though manuscripts deliver such valuable information, there are some things that need to be improved and addressed. Please see comments:

Line 20-21: "However, a vaccine simultaneously protects fish form these two bacterial infections was not yet available". Please edit to "However, a vaccine that simultaneously protects fish from these two bacterial infections was not yet available."

line 24: "their efficacy were evaluated" please change to "was evaluated"

line 29-30: "those from bivalent vaccine group were enhanced more 29 significantly (P < 0.05)". Please modify to "those from bivalent vaccine group were significantly enhanced (P < 0.05)"

Line 32: "highly efficacious" to "highly efficient"

Line 33: "divalent" to "bivalent"

Line 56 and 65: Nocardiosis or nocardiosis, please keep consistency.

Line 82: When referring to a " long-lasting immune response," you refer to an adaptive immune response; please use the correct terminology.

Line 84: "to protect economic fish" please change to "fish of economic importance."

Line 86: "and merely defend one pathogen disease". Please change to "and merely protect against a single pathogen disease."

line 144: "non-specific"

In section 2.6: Were serum samples previously inactivated and treated with chloroform prior ELISA proceeding ??? or is this information part of the citation 34?

In section 2.7: please provide the primer efficiency and the calculation method for it. The information requested is relevant, as the B-actin reference gene does not show consistent Ct values in different treatments. Could you explain the reason why the usage of only one reference gene? Is this based on another study? If primer efficiency was calculated for the sample obtained in this study, please provide the supplementary data.

Section 3.5: RPS values in the result section are clearly explained; however, when reading the abstract, these values do not match the results description. Are those in the abstract the highest values? I will suggest dividing the highlights in the abstract into two parts, either single infection or double infection, to avoid confusing the reader.

Author Response

Dear Editor,

Thank you and the reviewers for giving us so many comments on the manuscript vaccines-3815078, entitled “Efficacy of Whole-cell-based Mono- and Bi-valent Vaccines against Nocardia seriolae and Aeromonas veronii in Largemouth Bass, Micropterus salmoides”. These comments are all valuable and very helpful for revising and improving our paper, as well as the important guiding significance to our research. We made a careful revision in which we fully addressed the issues point by point raised from the reviewers’ comments. The responds to the reviewer’s comments are listed as follows. And we hope that the revised manuscript is acceptable for publication.

The manuscript focuses on fully filling the knowledge gap and the need for bivalent vaccines to protect against N. seriolae and A. veronii. Results are conclusive and well presented. The aim of the study is well addressed. The authors did a complete assessment of innate and adaptive immune responses. Even though manuscripts deliver such valuable information, there are some things that need to be improved and addressed. Please see comments:

Comment 1: Line 20-21: "However, a vaccine simultaneously protects fish form these two bacterial infections was not yet available". Please edit to "However, a vaccine that simultaneously protects fish from these two bacterial infections was not yet available."

Response: The sentence has been revised following your kind suggestion. Specific revision as follow:

Line 21-22: “However, a vaccine that simultaneously protects fish from these two bacterial infections was not yet available.”

Comment 2: line 24: "their efficacy were evaluated" please change to "was evaluated"

Response: Thanks for your valuable suggestion. The "their efficacy were evaluated" has been revised as " their efficacy was evaluated". Specific revision as follow:

Line 24: “and their efficacy was evaluated by challenge tests.”

Comment 3: line 29-30: "those from bivalent vaccine group were enhanced more 29 significantly (P < 0.05)". Please modify to "those from bivalent vaccine group were significantly enhanced (P < 0.05)"

Response: Thanks for your kindly suggestion. The "those from bivalent vaccine group were enhanced more significantly (P < 0.05)" has been replaced as "those from bivalent vaccine group were significantly enhanced (P < 0.05)". Specific revision as follow:

line 30-31: “those from bivalent vaccine group were significantly enhanced (P < 0.05).”

Comment 4: Line 32: "highly efficacious" to "highly efficient"

Response: Thanks for your kindly suggestion. The "highly efficacious" has been revised as "highly efficient". Specific revision as follow:

Line 31-32: “In the challenge test, both monovalent and bivalent vaccines were found to be highly efficient”

Comment 5: Line 33: "divalent" to "bivalent"

Response: The word “divalent” has been replaced with “bivalent”. Specific revision as follow:

Comment 6: Line 56 and 65: Nocardiosis or nocardiosis, please keep consistency.

Response: Sorry for the mistake. The nocardiosis has been unified. Specific revision as follow:

Line 57-59: “N. seriolae and virulent A. veronii are common bacterial pathogens within the aquaculture industry that can cause nocardiosis and enteric septicemia disease in largemouth bass, respectively”

Line 316-317: “Recently, outbreaks of nocardiosis and enteric septicemia disease result in high mortality in intensive farming of largemouth bass”

Comment 7: Line 82: When referring to a "long-lasting immune response," you refer to an adaptive immune response; please use the correct terminology.

Response: Thanks for your kindly suggestion. The "long-lasting immune response" has been revised as "long-lasting adaptive immune response". Specific revision as follow:

Line 84-85: “Furthermore, vaccine administrations assist in initiating a long-lasting adaptive immune response against microorganisms.”

Comment 8: Line 84: "to protect economic fish" please change to "fish of economic importance."

Response: Thanks for your kindly suggestion. The " to protect economic fish" has been revised as " fish of economic importance". Specific revision as follow:

Line 86-87: “Based on these advantages, some vaccines have been extensively investigated and even commercially used to protect fish of economic importance.”

Comment 9: Line 86: "and merely defend one pathogen disease". Please change to "and merely protect against a single pathogen disease."

Response: Thanks for your kindly suggestion. The "and merely defend one pathogen disease" has been revised as "and merely protect against a single pathogen disease.". Specific revision as follow:

Line 88-89: “Whereas, most of the vaccines that have been reported are monovalent formats and merely protect against a single pathogen disease.”

Comment 10: line 144: "non-specific"

Response: Thanks for your kindly suggestion. The word "nonspecific" has been revised as "non-specific". Specific revision as follow:

Line 147-148: “and then stored at −80 °C for future detection of specific antibody titers and non-specific enzyme activities.”

Comment 11: In section 2.6: Were serum samples previously inactivated and treated with chloroform prior ELISA proceeding ??? or is this information part of the citation 34?

Response: Thanks for your kindly suggestion. The serum samples were heat inactivated at 56 °C for 30 min prior to ELISA assay. The specific treatment has been supplemented. Specific revision as follow:

Line 156-157: “Serum samples were heat inactivated at 56 °C for 30 min and used for specific antibody titers against A. veronii and N. seriolae by ELISA as previous reports”.

Comment 12: In section 2.7: please provide the primer efficiency and the calculation method for it. The information requested is relevant, as the B-actin reference gene does not show consistent Ct values in different treatments. Could you explain the reason why the usage of only one reference gene? Is this based on another study? If primer efficiency was calculated for the sample obtained in this study, please provide the supplementary data.

Response: Thanks for your valuable suggestion. It’s important to detect the primer efficiency before qRT-PCR assay. The primers used in the present research was referred to references [38-40], in which their primer efficiency has been confirmed. Also, the afore mentioned references were both selected the B-actin as one reference gene. Moreover, the Amplicon size and gene Accession No. have been added in Table 1. Specific revision as follow:

Target genes

F: Primer sequence (5'-3')

R: Primer sequence (5'-3')

Amplicon size (bp)

Accession No.

References

TNFα

CTTCGTCTACAGCCAGGCATCG

TTTGGCACACCGACCTCACC

161

XM_038710731.1

[38]

IL-1β

CGTGACTGACAGCAAAAAGAGG

GATGCCCAGAGCCACAGTTC

166

XM_046034892.1

[38]

IL-8

CGTTGAACAGACTGGGAGAGATG

AGTGGGATGGCTTCATTATCTTGT

107

MW751832.1

[38]

IL-10

ACAACCAGTGCTGCCGTT

GCAGCGCTGTGTCTAAGTCA

117

XM_038696252.1

[39]

IFN-γ

TGCAGGCTCTCAAACACATC

TGTTTTCGGTCAGTGTGCTC

105

XM_038707474.1

[40]

CD4-1

GCTCCAGCGGGGAATAATTT

GCCAGGCAAGCTCAAAGTTA

73

XM_038711093.1

[40]

CD8-α

GGAAGGGGATCCTGTTGACA

CCAGCACTCGAAACCAGATG

74

XM_038696403.1

[40]

IgM

CTGGACCAGTCTCCCTCTGA

CGAGGTACTGAGTGCTGCTG

235

MZ396108.1

[40]

IgT

AAAGGAGATGGGAGTGAGCC

GTTGGGTCTTCTGTGGGGG

199

MZ388129.1

[40]

MHCI-α

GTGGTTCAACGTCAACATCG

ACCCAGACTTGTTCGGTGTC

198

XM_046058860.1

[40]

MHCII-α

GAGGACCTTGCTGTCATTGG

GCGTACCAAACCTCTTCACC

98

XM038696307

[40]

β-actin

CCACCACAGCCGAGAGGGAA

TCATGGTGGATGGGGCCAGG

303

MH018565.1

[40]

Comment 13: Section 3.5: RPS values in the result section are clearly explained; however, when reading the abstract, these values do not match the results description. Are those in the abstract the highest values? I will suggest dividing the highlights in the abstract into two parts, either single infection or double infection, to avoid confusing the reader.

Response: Thanks for your kindly suggestion. The RPS of both monovalent and bivalent against signal and dual infection with N. seriolae and A. veronii were added in the Abstract section. Specific revision as follow:

Line 32-36: The Monovalent Av and Monovalent Ns against A. veronii and N. seriolae showed 56.67% and 22.22% RPS, respectively. Moreover, Bivalent Av-Ns offer 33.33% and 76.67% RPS for single infection with N. seriolae or A. veronii, as well as provide 44.44% RPS for dual infection with combined N. seriolae and A. veronii.

Sincerely yours,

Qiushi Zhang1,2,#, Nengbin Zhu1,#, Ruiping Xu1, Eakapol Wangkahart3, Lin Zhang4, Lihe Liu1, Rui Wang1, Zhen Xu2, Weiguang Kong2,*, Hongsen Xu1,*

1 Hubei Key Laboratory of Animal Nutrition and Feed Science, School of Animal Science and Nutritional Engineering, Wuhan Polytechnic University, Wuhan, 430023, China

2 Key Laboratory of Breeding Biotechnology and Sustainable Aquaculture, Institute of Hydrobiology, Chinese Academy of Sciences, Wuhan, 430072, China

3 Laboratory of Fish Immunology and Nutrigenomics, Applied Animal and Aquatic Sciences Research Unit, Division of Fisheries, Faculty of Technology, Mahasarakham University, Khamriang Sub-District, Kantarawichai, Mahasarakham, 44150, Thailand

4 Yangtze River Fisheries Research Institute, Chinese Academy of Fishery Sciences, Wuhan, 430223, China

* Correspondence: Hubei Key Laboratory of Animal Nutrition and Feed Science, School of Animal Science and Nutritional Engineering, Wuhan Polytechnic University, Wuhan, 430023, China

* E-mail addresses: kongweiguang@ihb.ac.cn (W. Kong), Hsxu1989@163.com (H. Xu)

# The first two authors are equally contributed.

Reviewer 2 Report

Comments and Suggestions for Authors

Dear authors, It was a real pleasure to read your manuscript. Developing effective fish vaccines is rather important for the future of aquaculture.

My only problem with the content of your paper is the fact, that you have been mainly focusing on the primary imune response (1-6 weeks after vaccination). Unfortunately there is only one experiment (challenge), that can be regarded as a secondary response. In other words: you do not give much information on the required amount of time for optimal formation of immunological memory after vaccination. It is known, that the development of optimal immunological memory in fish can take a considerable amount of time in the order of 3 - 10 months.

Moreover, the initial antigen dose for vaccination can have a significant effect on memory formation. My suggestion is, that you pay some more attention on the importance of memory formation in the Discussion paragraph.

See also the following publications: C.Ma, J. Ye, S.L. Kaattari, Differential compartmentalization of memory B cells versus plasma cells in salmonid fish, European Journal of Immunology 43 (2013) 360-370. J.O. Sunyer, P. Boudinot, B-cell responses and antibody repertoires in Teleost fish: From ag receptor diversity to immune memory and vaccine development. In: K. Buchmann, C.J. Secombes (Eds.) Springer Publ. (2022) 253-278.

Author Response

Dear Editor,

Thank you and the reviewers for giving us so many comments on the manuscript vaccines-3815078, entitled “Efficacy of Whole-cell-based Mono- and Bi-valent Vaccines against Nocardia seriolae and Aeromonas veronii in Largemouth Bass, Micropterus salmoides”. These comments are all valuable and very helpful for revising and improving our paper, as well as the important guiding significance to our research. We made a careful revision in which we fully addressed the issues point by point raised from the reviewers’ comments. The responds to the reviewer’s comments are listed as follows. And we hope that the revised manuscript is acceptable for publication.

Dear authors, It was a real pleasure to read your manuscript. Developing effective fish vaccines is rather important for the future of aquaculture.

Comment 1: My only problem with the content of your paper is the fact, that you have been mainly focusing on the primary immune response (1-6 weeks after vaccination). Unfortunately, there is only one experiment (challenge), that can be regarded as a secondary response. In other words: you do not give much information on the required amount of time for optimal formation of immunological memory after vaccination. It is known, that the development of optimal immunological memory in fish can take a considerable amount of time in the order of 3 - 10 months. Moreover, the initial antigen dose for vaccination can have a significant effect on memory formation. My suggestion is, that you pay some more attention on the importance of memory formation in the Discussion paragraph.

Response: Thanks for your kind suggestion. The importance of memory formation after vaccination was discussed in the Discussion Section. Moreover, your suggestion provide idea for our subsequent research. In our future research, we will pay attention to the amount of time for optimal formation of immunological memory after vaccination. Specific revision as follow:

Line 390-411: It’s well known that successful vaccination depends on the ability of improving the immunological memory to respond swiftly and effectively towards a subsequent infection by the same specific microbes [56]. This can be accomplished by the previous production of long-lived memory T and B lymphocytes during the clonal expansion of antigen-specific cells in the primary response [57]. In the present research, we immunized largemouth bass at one time and then challenged them with related bacteria at 6th wpv. The results showed that high protection was obtained in vaccinated fish in compare to those from control group. This is perhaps due to the production of immunological memory that help fish defend against a subsequent re-encounter with the targeted pathogen. Previous research has found that the antibody levels of fish in which immunized with inactivated vaccine for 8 weeks were increased immediately after challenging within 1 week [58]. Many factors, such as vaccination method, initial antigen dose and antigen types, that may affect the formation of immune memory [59]. The tilapia intraperitoneally immunized with an inactivated vaccines showed a significant RPS than those vaccinated via intramuscular route [60]. This difference was due to the fact that bioavailability of the antigens via intraperitoneal route was 4.5 times higher than via intramuscular route [61]. The dose dependency of the memory formation was studied in detail in carp, and found that fish immunized with 109 sheep red blood cells induce high levels of secondary response [62]. Previous research showed that specificity and immunological memory were activated by Edwardsiella tarda vaccine (109 CFU/mL) as demonstrated by the increase of antibody titer and percentage of CD4+ T and CD8+ T lymphocytes [63]. In the present research, 109 CFU/mL formalin killed bacteria were selected as target vaccine for immunization.

Supplemented Reference

[56] T. Yamaguchi, E. Quillet, P. Boudinot, U. Fischer, What could be the mechanisms of immunological memory in fish?, Fish & Shellfish Immunology 85 (2019) 3-8.

[57] C. Ma, J. Ye, S.L. Kaattari, Differential compartmentalization of memory B cells versus plasma cells in salmonid fish, European Journal of Immunology 43(2) (2013) 360-370.

[58] P. Tanpichai, S. Chaweepack, S. Senapin, P. Piamsomboon, J. Wongtavatchai, Immune activation following vaccination of Streptococcus iniae bacterin in Asian seabass (Lates calcarifer, Bloch 1790), Vaccines 11(2) (2023) 351.

[59] J.O. Sunyer, P. Boudinot, B-cell responses and antibody repertoires in teleost fish: From Ag receptor diversity to immune memory and vaccine development, Principles of Fish Immunology: From Cells and Molecules to Host Protection (2022) 253-278.

[60] P.H. Klesius, C.A. Shoemaker, J.J. Evans, Efficacy of single and combined Streptococcus iniae isolate vaccine administered by intraperitoneal and intramuscular routes in tilapia (Oreochromis niloticus), Aquaculture 188(3-4) (2000) 237-246.

[61] A. Karami, A. Christianus, Z. Ishak, M.A. Syed, S.C. Courtenay, The effects of intramuscular and intraperitoneal injections of benzo [a] pyrene on selected biomarkers in Clarias gariepinus, Ecotoxicology and environmental safety 74(6) (2011) 1558-1566.

[62] G. Rijkers, E. Frederix-Wolters, W.B. van MUISWINKEL, The immune system of cyprinid fish. The effect of antigen dose and route of administration on the development of immunological memory in carp (Cyprinus carpio), Phylogeny of immunological memory, North-Holland Publishing Company1980, pp. 93-102.

[63] X. Wu, J. Xing, X. Tang, X. Sheng, H. Chi, W. Zhan, Protective cellular and humoral immune responses to Edwardsiella tarda in flounder (Paralichthys olivaceus) immunized by an inactivated vaccine, Molecular Immunology 149 (2022) 77-86.

Sincerely yours,

Qiushi Zhang1,2,#, Nengbin Zhu1,#, Ruiping Xu1, Eakapol Wangkahart3, Lin Zhang4, Lihe Liu1, Rui Wang1, Zhen Xu2, Weiguang Kong2,*, Hongsen Xu1,*

1 Hubei Key Laboratory of Animal Nutrition and Feed Science, School of Animal Science and Nutritional Engineering, Wuhan Polytechnic University, Wuhan, 430023, China

2 Key Laboratory of Breeding Biotechnology and Sustainable Aquaculture, Institute of Hydrobiology, Chinese Academy of Sciences, Wuhan, 430072, China

3 Laboratory of Fish Immunology and Nutrigenomics, Applied Animal and Aquatic Sciences Research Unit, Division of Fisheries, Faculty of Technology, Mahasarakham University, Khamriang Sub-District, Kantarawichai, Mahasarakham, 44150, Thailand

4 Yangtze River Fisheries Research Institute, Chinese Academy of Fishery Sciences, Wuhan, 430223, China

* Correspondence: Hubei Key Laboratory of Animal Nutrition and Feed Science, School of Animal Science and Nutritional Engineering, Wuhan Polytechnic University, Wuhan, 430023, China

* E-mail addresses: kongweiguang@ihb.ac.cn (W. Kong), Hsxu1989@163.com (H. Xu)

# The first two authors are equally contributed.

Reviewer 3 Report

Comments and Suggestions for Authors

Comments on the manuscript with ID (vaccines-3815078-peer-review-v1).

Line 149: Add codes of the kits used.

Line 183: Add a suitable reference for the methodology used.

Livak, K. J., & Schmittgen, T. D. (2001). Analysis of relative gene expression data using real-time quantitative PCR and the 2− ΔΔCT method. Methods, 25(4), 402-408.‏

Line 183: Why did the authors select β-actin as a housekeeping gene?

Table 1: This table missed important information such as NCBI GenBank accession numbers, R2, Pearson’s coefficient, efficiency and Annealing temperatures for each gene. You should also write the abbreviations of each gene below the table.

Line 190: Did you perform LD50 testing before selection of the bacterial infection dose? Data was not presented.

References

  • Latin names should be written in italics (see the attached PDF file).
  • Revise the Journal names (see the attached PDF file).
  • There is a mistake in reference (37).

Author Response

Dear Editor,

Thank you and the reviewers for giving us so many comments on the manuscript vaccines-3815078, entitled “Efficacy of Whole-cell-based Mono- and Bi-valent Vaccines against Nocardia seriolae and Aeromonas veronii in Largemouth Bass, Micropterus salmoides”. These comments are all valuable and very helpful for revising and improving our paper, as well as the important guiding significance to our research. We made a careful revision in which we fully addressed the issues point by point raised from the reviewers’ comments. The responds to the reviewer’s comments are listed as follows. And we hope that the revised manuscript is acceptable for publication.

Comment 1: Line 149: Add codes of the kits used.

Response: Thanks for your kindly suggestion. The codes of all kits have been added. Specific revision as follow:

Line 151-154: After thawing, the serum samples were selected to determine activities of acid phos-phatase (ACP; Cat. No. A060–2), alkaline phosphatase (AKP; Cat. No. A059–2) and lyso-zyme (LZM; Cat. No. A050) as per the manufacturer’s instructions of commercial kits produced by Nanjing Jiancheng Bioengineer Institute (Jiangsu, China).

Comment 2: Line 183: Add a suitable reference for the methodology used. Livak, K. J., & Schmittgen, T. D. (2001). Analysis of relative gene expression data using real-time quantitative PCR and the 2− ΔΔCT method. Methods, 25(4), 402-408.‏

Response: Thanks for your kindly suggestion. The reference has been added. Specific revision as follow:

Line 186-187: The relative gene expression values were analyzed using the 2-ΔΔCT method [37]

Reference:

[37] K.J. Livak, T.D. Schmittgen, Analysis of relative gene expression data using real-time quantitative PCR and the 2−ΔΔCT method, methods 25(4) (2001) 402-408.

Comment 3: Line 183: Why did the authors select β-actin as a housekeeping gene?

Response: Thanks for your kindly suggestion. In the present research, we select β-actin as a housekeeping gene referred to references [38-40].

Line 187: and β-actin was used as the housekeeping gene following previous researches [38-40]

Comment 4: Table 1: This table missed important information such as NCBI GenBank accession numbers, R2, Pearson’s coefficient, efficiency and Annealing temperatures for each gene. You should also write the abbreviations of each gene below the table.

Response: Thanks for your valuable suggestion. It’s important to detect the primer efficiency before qRT-PCR assay. The primers used in the present research was referred to references [38-40], in which their primer efficiency has been confirmed. Also, the afore mentioned references were both selected the B-actin as one reference gene. Moreover, the Amplicon size and gene Accession No. have been added in Table 1. Specific revision as follow:

Target genes

F: Primer sequence (5'-3')

R: Primer sequence (5'-3')

Amplicon size (bp)

Accession No.

References

TNFα

CTTCGTCTACAGCCAGGCATCG

TTTGGCACACCGACCTCACC

161

XM_038710731.1

[38]

IL-1β

CGTGACTGACAGCAAAAAGAGG

GATGCCCAGAGCCACAGTTC

166

XM_046034892.1

[38]

IL-8

CGTTGAACAGACTGGGAGAGATG

AGTGGGATGGCTTCATTATCTTGT

107

MW751832.1

[38]

IL-10

ACAACCAGTGCTGCCGTT

GCAGCGCTGTGTCTAAGTCA

117

XM_038696252.1

[39]

IFN-γ

TGCAGGCTCTCAAACACATC

TGTTTTCGGTCAGTGTGCTC

105

XM_038707474.1

[40]

CD4-1

GCTCCAGCGGGGAATAATTT

GCCAGGCAAGCTCAAAGTTA

73

XM_038711093.1

[40]

CD8-α

GGAAGGGGATCCTGTTGACA

CCAGCACTCGAAACCAGATG

74

XM_038696403.1

[40]

IgM

CTGGACCAGTCTCCCTCTGA

CGAGGTACTGAGTGCTGCTG

235

MZ396108.1

[40]

IgT

AAAGGAGATGGGAGTGAGCC

GTTGGGTCTTCTGTGGGGG

199

MZ388129.1

[40]

MHCI-α

GTGGTTCAACGTCAACATCG

ACCCAGACTTGTTCGGTGTC

198

XM_046058860.1

[40]

MHCII-α

GAGGACCTTGCTGTCATTGG

GCGTACCAAACCTCTTCACC

98

XM038696307

[40]

β-actin

CCACCACAGCCGAGAGGGAA

TCATGGTGGATGGGGCCAGG

303

MH018565.1

[40]

Comment 5: Line 190: Did you perform LD50 testing before selection of the bacterial infection dose? Data was not presented.

Response: Thanks for your kindly question. The LD50 testing of each bacterium has been done in our previous research. Please see reference 26

Reference:

[26] H. Xu, R. Xu, X. Wang, Q. Liang, L. Zhang, J. Liu, J. Wei, Y. Lu, D. Yu, Co-infections of Aeromonas veronii and Nocardia seriolae in largemouth bass (Micropterus salmoides), Microbial Pathogenesis 173 (2022) 105815.

Comment 6: References

  • Latin names should be written in italics (see the attached PDF file).
  • Revise the Journal names (see the attached PDF file).
  • There is a mistake in reference (37).

Response: Thanks for your kindly suggestion. We have comparatively revised the Reference and corrected the reference 41.

Sincerely yours,

Qiushi Zhang1,2,#, Nengbin Zhu1,#, Ruiping Xu1, Eakapol Wangkahart3, Lin Zhang4, Lihe Liu1, Rui Wang1, Zhen Xu2, Weiguang Kong2,*, Hongsen Xu1,*

1 Hubei Key Laboratory of Animal Nutrition and Feed Science, School of Animal Science and Nutritional Engineering, Wuhan Polytechnic University, Wuhan, 430023, China

2 Key Laboratory of Breeding Biotechnology and Sustainable Aquaculture, Institute of Hydrobiology, Chinese Academy of Sciences, Wuhan, 430072, China

3 Laboratory of Fish Immunology and Nutrigenomics, Applied Animal and Aquatic Sciences Research Unit, Division of Fisheries, Faculty of Technology, Mahasarakham University, Khamriang Sub-District, Kantarawichai, Mahasarakham, 44150, Thailand

4 Yangtze River Fisheries Research Institute, Chinese Academy of Fishery Sciences, Wuhan, 430223, China

* Correspondence: Hubei Key Laboratory of Animal Nutrition and Feed Science, School of Animal Science and Nutritional Engineering, Wuhan Polytechnic University, Wuhan, 430023, China

* E-mail addresses: kongweiguang@ihb.ac.cn (W. Kong), Hsxu1989@163.com (H. Xu)

# The first two authors are equally contributed.

Round 2

Reviewer 1 Report

Comments and Suggestions for Authors

I want ot congratualte the authors for their detailed report on the suggestions provided and to address every comment and previous concerns

Author Response

Thanks for your recognition!

Reviewer 3 Report

Comments and Suggestions for Authors

The authors appropriately responded to the points and comments raised by the reviewer. However, i found that the iThenticate report = 39 % similarity especially in the result section.

The authors should work to decrease this percentage to avoid pelagirism with the published data.

Author Response

Dear editor

We have already reduced the repetition in the MS, and hope it could be acceptable in the present revision.